# Atomic model of vesicular stomatitis virus and mechanism of assembly

Kang Zhou[1,2,3,7], Zhu Si[1,2,7], Peng Ge [2,6], Jun Tsao[4], Ming Luo [5] & Z. Hong Zhou [1,2] ✉

Like other negative-strand RNA viruses (NSVs) such as influenza and rabies, vesicular stomatitis virus (VSV) has a three-layered organization: a layer of matrix protein (M) resides between the glycoprotein (G)-studded membrane envelope and the nucleocapsid, which is composed of the nucleocapsid protein (N) and the encapsidated genomic RNA. Lack of in situ atomic structures of these viral components has limited mechanistic understanding of assembling the bullet-shaped virion. Here, by cryoEM and sub-particle reconstruction, we have determined the in situ structures of M and N inside VSV at 3.47 Å resolution. In the virion, N and M sites have a stoichiometry of 1:2. The in situ structures of both N and M differ from their crystal structures in their N-terminal segments and oligomerization loops. N-RNA, N-N, and N-M-M interactions govern the formation of the capsid. A double layer of M contributes to packaging of the helical nucleocapsid: the inner M (IM) joins neighboring turns of the N helix, while the outer M (OM) contacts G and the membrane envelope. The pseudo-crystalline organization of G is further mapped by cryoET. The mechanism of VSV assembly is delineated by the network interactions of these viral components.

Negative-strand RNA viruses (NSVs) constituting the *Haploviricotina* subphylum of the *Negarnaviricota* phylum include some of the most devastating human pathogens, such as rabies virus (RABV) (*Rhabdoviridae*), measles virus (MeV) (*Paramyxoviridae*), influenza virus (*Orthomyxoviridae*), and Marburg and Ebola viruses (*Filoviridae*). Vesicular stomatitis virus (VSV) is an enveloped, bullet-shaped rhabdovirus, which is closely related to the highly contagious and historically significant RABV[1,2]. VSV is the prototypical NSV and has long been used as a model for RABV and other NSVs. It has also been widely used for engineering pseudotypes—VSV-like particles carrying effector molecules—as both vaccines (including those against SARS-CoV-2, the virus responsible for the COVID-19 pandemic[3,4]) and anti-cancer agents[5]. Attenuated VSV strains are non-toxic to normal tissues, as they effectively trigger an interferon-mediated anti-viral response

against themselves[6]. Vaccines based on pseudo-type VSV carrying foreign surface proteins or recombinant VSV expressing foreign viral genes have been developed as vaccine candidates for COVID-19 (ref. 7), human immuno-deficiency virus (HIV)[8,9], "avian flu"[10], measles[11], Ebola virus[12] and Marburg virus[12,13]. Pseudotypes of VSV that carry HIV receptors selectively target and kill HIV-1 infected cells and control HIV-1 infection[14,15]. The atomic details of the molecular interactions governing VSV assembly would benefit all these therapeutic and pro-phylactic endeavors.

All members of *Rhabdoviridae* encode three structural proteins: nucleocapsid protein (N), matrix protein (M) and glycoprotein (G). In the virion, these proteins are organized into a distinctive bullet shape. M is sandwiched between a G-containing membrane envelope and a nucleocapsid, composed of N and a

[1]Department of Microbiology, Immunology & Molecular Genetics, University of California, Los Angeles (UCLA), Los Angeles, CA 90095, USA. [2]California NanoSystems Institute, UCLA, Los Angeles, CA 90095, USA. [3]School of Life Sciences, University of Science and Technology of China, Hefei, Anhui 230026, P. R. China. [4]Department of Microbiology, University of Alabama at Birmingham, Birmingham, Al 35294, USA. [5]The Department of Chemistry, Georgia State University, Atlanta, GA 30303, USA. [6]Present address: Departments of Chemistry and Biochemistry and Biological Chemistry, and Howard Hughes Medical Institute, UCLA, Los Angeles, CA 90095, USA. [7]These authors contributed equally: Kang Zhou, Zhu Si. ✉e-mail: Hong.Zhou@UCLA.edu

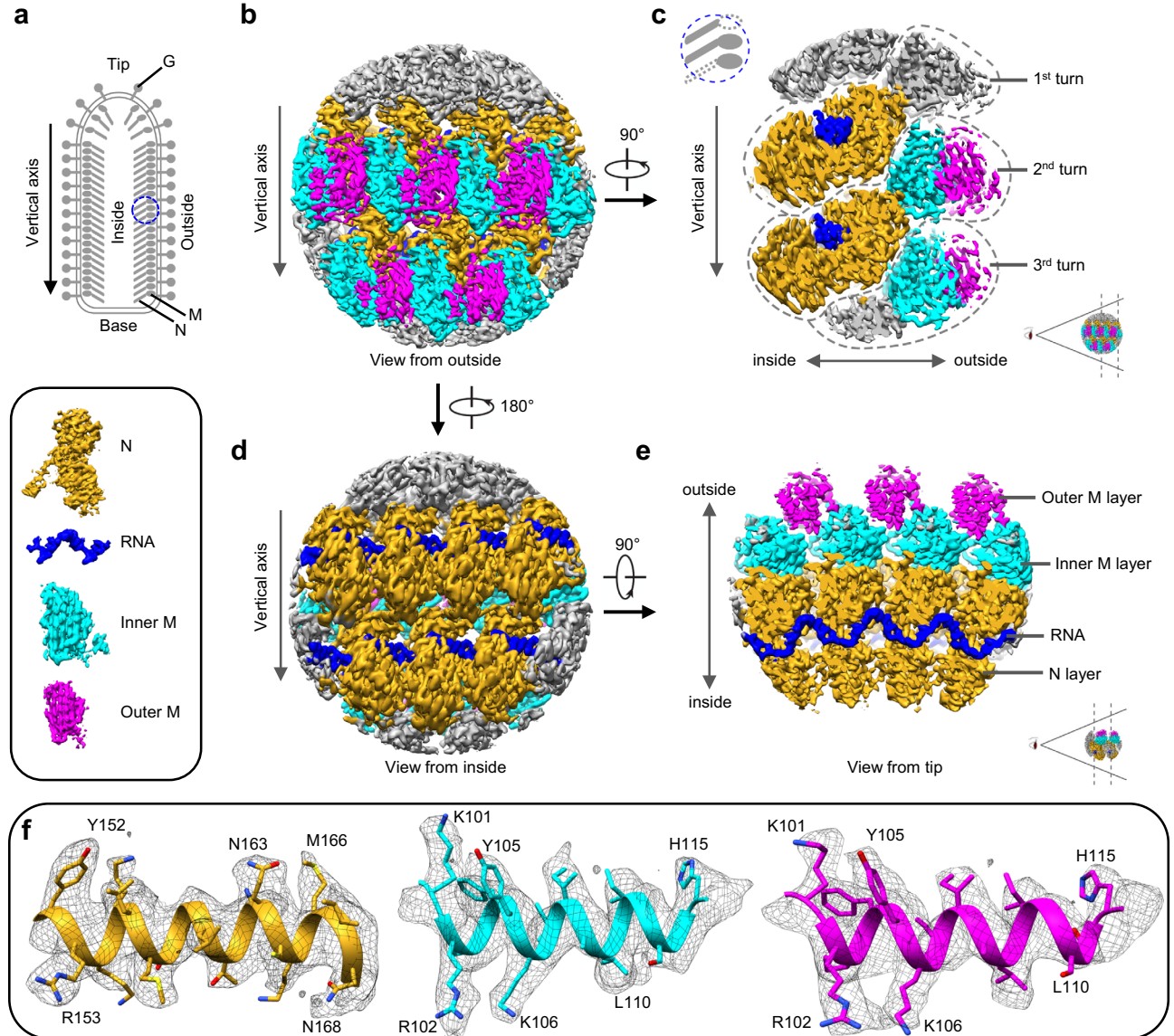

**Fig. 1 | Sub-particle reconstruction of VSV trunk at near-atomic resolution.**
**a** Schematic diagram of the VSV virion. Blue circle marks the region used for sub-particle reconstruction. **b**, **d** CryoEM density maps of partial VSV nucleocapsid shown in two opposite views. Only intact N, intact M and RNA are colored for clarity; other densities are gray. N subunits are colored in goldenrod, inner M in cyan, outer M in magenta and RNA in blue (inset). **c**, **e** Cross sections of cryoEM density maps viewed from side and tip. **f** CryoEM density maps (mesh) of α-helices from three kinds of subunits.

single-stranded genomic RNA. The crystal structures have been reported for the C-terminal core domain of M (*i.e.*, $M^t$)[16], N[17] and the two forms of the ectodomain of G[18,19]. Previous cryo electron microscopy (cryoEM) structures of VSV from helical reconstruction at 10.6 Å resolution have led to a model of 3′ to 5′ RNA-guided assembly of the nucleocapsid from the tip of the bullet to the trunk, forming a bullet-shaped viral particle[20]. However, without in situ structures of M and N in the virion at a near-atomic resolution, the precise arrangement of M, N and G and their interactions governing the virion assembly remain elusive, limiting our understanding of VSV assembly and impeding rational engineering efforts of VSV pseudotypes.

In this study, we determined the in situ structures of M and N inside VSV at 3.47 Å resolution by cryoEM and sub-particle reconstruction. The structure shows that each N corresponds to a pair of M subunits sites in the virion, not the previously reported single M. Cryo electron tomography (cryoET) and sub-tomogram averaging further establish the pseudo-crystalline organization of G with predominantly hexagonal and occasionally pentagonal tiles on the membrane envelope. The interactions between N and M, as well as matching distributions of M and G, suggest a mechanism of VSV assembly.

## Results

### In situ structures of M, N and encapsidated RNA resolved at near atomic resolution by helical sub-particle reconstruction

The large variable range of subunits/turn (35.5–41.5 subunits/turn, though previous studies have shown mostly 37.5 subunits[20]) has hitherto presented challenges in reconstructing the VSV trunk to a resolution needed for atomic modeling using the conventional helical reconstruction method. Here, we were able to determine the in situ structure of the VSV trunk to 3.47 Å resolution (Fig. 1 and Supplementary Video 1) by subjecting particles to fine 3D classification based on subunits/turn and then treating small regions in the major class (38.5 subunits/turn) independently, a novel approach which we termed helical sub-particle reconstruction (see Method for details). The structure reveals the 1:2 ratio between N and M sites and resolves amino acid side

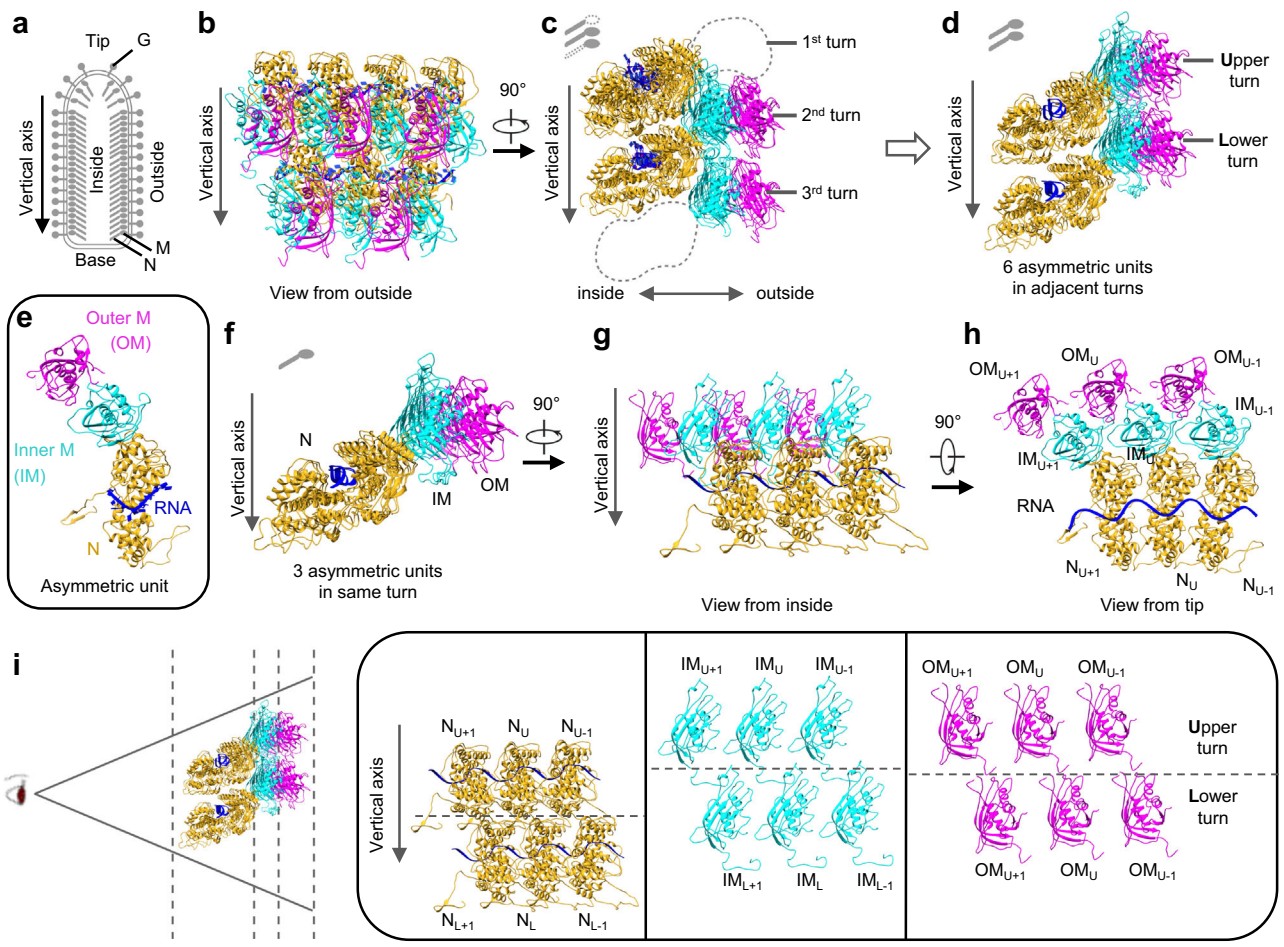

**Fig. 2 | Atomic model of the partial VSV nucleocapsid. a** Schematic diagram of the VSV virion. **b**, **c** Two orthogonal views of the atomic model, which contains 7 N (goldenrod), 7 inner M (cyan) and 5 outer M (magenta). Only intact N, intact M and RNA are built and colored, as in the cryoEM map. **d** Six asymmetric units from two turns derived from (**c**). **e** One asymmetric unit, including 1 N, 1 IM and 1 OM, together with 9 RNA nucleotides. **f–h** Three asymmetric units from the same turn shown in three orthometric views. **i** Layer expansion of (**d**) from the central axis of the virion. Each subunit is labeled based on its lateral relationship (+1 or −1, along the left-handed helix) and vertical position (Upper (U) and Lower (L) turns).

chains and RNA bases, both of which were used to build the atomic models (Fig. 1c and Supplementary Video 1).

We were able to fit an atomic model of 19 protein subunits, comprising 7 N and 12 M, and two long segments of single stranded RNA (33 and 35 nucleotides in length, respectively) into our density map obtained by sub-particle reconstruction (Fig. 2b, c and Supplementary Video 2). Based on the extent of interactions between subunits, we defined an asymmetric unit to contain an N subunit and two M subunits (Fig. 2e). The atomic model contains 19 subunits arranged across three radial layers from inner to outer: N layer, inner M (IM) layer, and outer M (OM) layer in one intact and two partial helical turns of the asymmetric units along the helical axis (Fig. 2c). Within the same turn, each IM subunit interacts with two N subunits, one N from the same asymmetric unit and another from a neighboring asymmetric unit (Fig. 2f–h). This IM subunit is also inlaid between two OM subunits, one OM from the same asymmetric unit and another from a neighboring asymmetric unit (Fig. 2h). Each IM subunit is placed between N subunits from two successive (upper-lower) turns (Fig. 2d), but the IM subunit in lower turn has no significant interactions with the N subunit from upper turn. In addition to the close lateral interactions between two adjacent N subunits in the same turn, the single-stranded RNA threads through the cleft of N subunits (Fig. 2h). Each N subunit accommodates 9 nucleotides of RNA (Fig. 2e).

## Comparison between in situ and crystal structures of N

The in situ atomic model of the full-length N (Fig. 3a) exhibits significant differences from the crystal structure (PDB: 2GIC)[17] of recombinantly-expressed N. The crystal structure comprises 10 subunits in one turn, or a decamer, while each turn in the in situ VSV trunk ranges from 35.5 to 41.5 subunits. When the in situ and crystal structures of single N subunit are compared, the main body and RNA-binding pattern remain the same between two structures, but there are significant differences in the N-terminal arm (Ser2 – Val25, termed N-arm) and an extended loop (Leu344 – Tyr355, termed C-loop) (Fig. 3a). Compared to those in the crystal structure (blue), in in situ N (red), the N-arm rotates downwards (towards the base) by 30 degrees, and the C-loop rotates outwards (towards the outside of the capsid) by 13 degrees (Fig. 3a). Superposition of the crystal and in situ structures of three adjacent N subunits was then performed by aligning only the middle N subunits ($N_U$) (Fig. 3b) while allowing the adjacent subunits to be not aligned. Interestingly, a stable interlocking anchor is found in the interactions among the three N subunits in both the crystal and the in situ structures (Fig. 3b and inset). The N-arm from the first N subunit ($N_{U-1}$) forms a β-hairpin and extends along the edge of $N_U$. On the opposite side, the C-loop from the third N subunit ($N_{U+1}$) runs over the C-domain of $N_U$ and doubles back. From the outside view of the nucleocapsid, the N-arm from the $N_{U-1}$ subunit and the C-loop from the

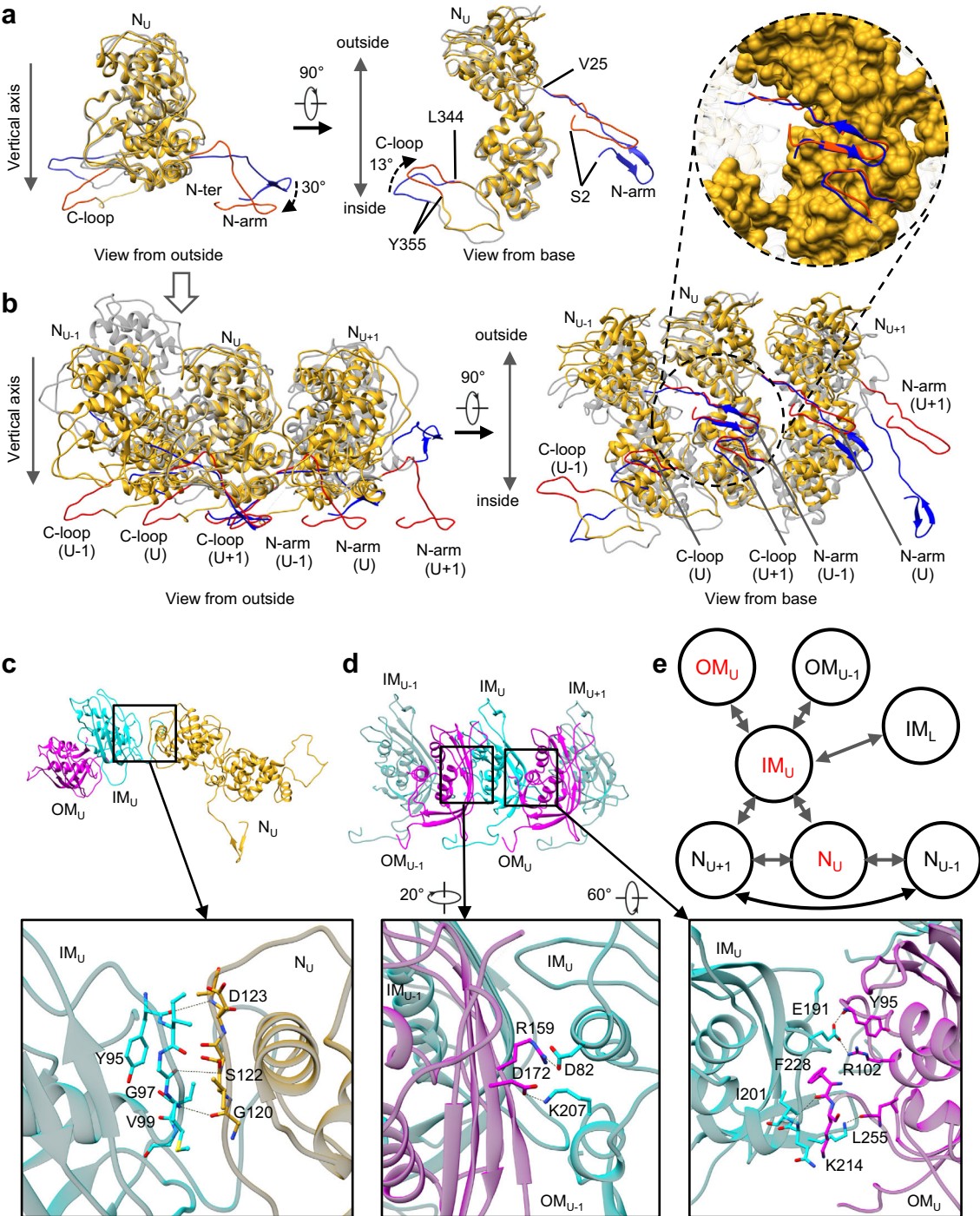

**Fig. 3 | Analysis and illustration of subunits interactions. a** Superposition of the in situ and crystal structures of a single N subunit. The in situ structure is colored in goldenrod except for the N-arm and C-loop, in red, while the crystal structure is colored in transparent gray except for the N-arm and C-loop, in blue.
**b** Superposition of three N subunits extended from (**a**). Only the middle N subunits are aligned. Inset: $N_U$ is shown as surface contour. **c** Interactions between IM and N from the same asymmetric unit. **d** Interactions between IM and two OMs, one OM from the same asymmetric unit and another one from the adjacent asymmetric unit. **e** The subunit interaction network in the VSV nucleocapsid, in which IM plays a key mediating role. As labeled, subunits $N_U$, $IM_U$ and $OM_U$ belong to the same asymmetric unit.

$N_{U+1}$ subunit form a "handhold", which holds in place the $N_U$ subunit (Fig. 3b, left); the interactions and relative position between the "handhold" and the $N_U$ are constant across the turn (Fig. 3b and inset). In this architecture, one N subunit connects its N-arm and C-loop to the N that are two subunits away on either side ($N_{U-2}$, $N_{U+2}$). Adjusting the conformation of the N-arm and C-loop allows N to adopt to different subunit numbers per turn with different diameters. In the in situ

structure, the N-arm and C-loop pivot outwards, which makes the ring larger to accommodate more subunits per turn.

## Comparison between in situ and crystal structures of M and between IM and OM

The final sub-particle reconstruction clearly shows that there is a double layer of M surrounding the helical nucleocapsid in the virion

(Figs. 1c and 2c). The IM layer bridges between the N layer and the OM layer; there is no direct contact between N and OM (Fig. 2c, d, f). The distances between the lateral IM subunits, as well as between the lateral OM subunits, are too large to establish significant IM-IM or OM-OM interactions (Fig. 2h, i).

The crystal structure of M[21] (PDB: 2W2R) was fitted into the corresponding EM densities. The main body of the crystal structure fit well, but an N-terminal segment of 12 residues (Leu41 – Asp52) could not fit. A segment of five residues (Leu53 – Asp57) is missing in the crystal structure of M (Supplementary Fig. 1a); however, the distance between Asp52 and Ser58 in the EM density is 46.6 Å (Supplementary Fig. 1b), too large to be spanned by five amino acid residues even when the main-chain is fully extended. Thus, the 12-residue segment and the main body next to each other cannot belong to the same M subunit in the in situ structure. In the crystal structure of M, the 12-residue segment could be assigned to another M molecule in the neighboring asymmetric unit[21]. After examining the EM densities and locations of M subunits, we focused on a pair of IM subunits, one from the upper helical turn (IM$_U$) and the other from the lower turn (IM$_L$) (Fig. 2i and Supplementary Fig. 1b). The distance between Asp52 (next to IM$_L$) and Ser58 (IM$_U$) is only 10.5 Å (Supplementary Fig. 1b), which is much shorter than the previously mentioned distance of 46.6 Å and within the range for inserting five residues. Based on this observation, the 12-residue segment (Leu41 – Asp52, beside IM$_L$) was assigned to IM$_U$, and five residues were added to that segment after Asp52 (Leu53 – Asp57) (Supplementary Fig. 1a, b). We also observed that the 12-residue segment interacts with IM$_L$ (Supplementary Fig. 1b). This completes an in situ atomic model of IM from residue Leu41 to the last residue at the C terminal end (Supplementary Fig. 1a), but the N-terminal 40 residues (Met1 – Pro40) remain disordered.

Superposition of IM and OM reveals three differences between the two molecules (Supplementary Fig. 1c). First, the 12-residue segment (Leu41 – Asp50) is unresolved in OM, suggesting flexibility. Second, residues Val122 – Asn127 in OM are also unresolved, and the segment from Asn194 to Ser199 in OM has a conformation different from that in IM (Supplementary Fig. 1c). All three conformational differences of OM are located at its outward-facing regions, far from N and IM layers but next to the viral envelope. Third, IM and OM are oriented completely differently inside the virion (Fig. 2i). These structural differences between IM and OM suggest that IM and OM play different roles in capsid assembly.

## Subunit interactions governing virion assembly

To understand VSV virion assembly, we used the PISA software[22] to calculate the Gibbs free energy and the interface area between subunits in order to quantify inter-subunit interactions in the nucleocapsid (Supplementary Table 1). The lateral interactions of adjacent N subunits are the strongest, while those of N$_{U-1}$-N$_{U+1}$ subunits are weaker but still significant (Fig. 3b and Supplementary Table 1, rows in blue). Remarkably, although N subunits from successive turns have a buried interface, there are no significant vertical interactions between the N subunits (Fig. 2d and Supplementary Table 1, third and fourth rows).

Despite its location between two successive turns of N subunits (Fig. 2d), the IM subunit interacts significantly only with two N subunits in the same turn, not the N in any other turns (Fig. 2h, i and Supplementary Table 1, rows in orange). These interactions occur on the inner side of the IM. The residues D123/S122/G120 from N$_U$ and Y95/G97/V99 from IM$_U$ comprise the main interactions between the two (Fig. 3c). There is a buried interface between IM$_L$ and N$_U$, but there are no strong interactions between them (Fig. 2i and Supplementary Table 1, fifth row). On its outer side, the IM subunit interacts with two independent OM subunits (Fig. 2h). One OM subunit is in the same asymmetric unit as the IM subunit and binds to that IM subunit more strongly than the other OM subunit from the neighboring asymmetric unit (Fig. 2h and Supplementary Table 1, rows in green); residues involved include E191/

F228/I201/D82/K207 from IM$_U$, Y95/R102/L255/K214 from OM$_U$ and R159/D172 from OM$_{U-1}$ (Fig. 3d).

In general, the interactions among subunits can be classified into three types according to their directions: lateral, radial and vertical (Fig. 3e). The lateral interactions are mainly at the N-N interfaces (Fig. 3b), including those between N$_U$ and N$_{U+1}$ and between N$_{U-1}$ and N$_{U+1}$. The lateral interactions are enhanced by the single-stranded RNA woven through the N subunits (Fig. 2h). The radial interactions are mediated by IM subunits: one IM subunit interacts with two N subunit inwards and with two OM subunits outwards (Fig. 2h). The main vertical interactions are between the N-terminal segment (Leu41 – Met48) of IM$_U$ and the main body of IM$_L$ (Fig. 2i and Supplementary Fig. 1b), and those interactions determine the distance between successive turns (Supplementary Fig. 1b). These three types of interactions govern the structure of the virion but do not explain the attachments of G.

## Local clustering of G trimers on the virion surface

To explore the overall organization of N, M and G and the interactions between them, we recorded cryoET tilt series from VSV samples with and without a density gradient step. Both sets of reconstructed tomograms (Fig. 4a, e and Supplementary Video 3) resolve the 3D organizations of VSV virions: the regularly patterned M and N proteins inside the virion are clearly visible in the tomograms, with G existing primarily as trimers on the viral membrane envelope but in different conformations (Fig. 4a–c, e–g). Although both samples were under neutral pH condition (pH = 7.4), the G trimers on the virions purified without a density gradient step are predominantly in the prefusion conformation[19] (Fig. 4a–c), which are shorter and wider than those in the postfusion conformation, as seen on the virions purified with a density gradient centrifugation step (Fig. 4e–g). In the reconstructed tomograms of both types of virions, we observed local G trimer clusters on the viral envelope arranged in hexagon tiles, and occasionally in pentagon tiles in the tip area of the virion (Fig. 4d, h). The observed clustering of multiple trimers is relevant to the virus-cell fusion event, which presumably requires concerted participation of multiple trimers to create a pore of fused virus-cell membranes.

By averaging a total of 4452 subtomograms of prefusion G trimers and 6030 subtomograms of postfusion G trimers, in situ structures of the G trimer in the prefusion (Fig. 5a) and postfusion (Fig. 5b) conformations were obtained at a resolution of 15.8 Å and 14.9 Å, respectively (Supplementary Fig. 2c). Unlike that of the prefusion conformation, the subtomogram average of the postfusion conformation shows that the central G trimer is surrounded by three blurred trimeric density blobs of similar size, each contacting one protomer (i.e., a G monomeric subunit) of the central G trimer (Fig. 5b, second row). The smeared density appearance of the surrounding trimeric blobs suggests partial occupancy of the surrounding trimer sites. Therefore, we performed subsequent symmetry relaxation and refinement, which yielded four different categories of G trimer supercomplexes, all containing a central but different numbers of neighboring G trimers (Fig. 5c). Among the four types of super-complexes, the super-complex containing one neighboring G trimer has the largest number of subtomograms (42.8% of the total subtomograms), followed by ones with two neighboring (33.2%), none neighboring (15.2%) and three neighboring (8.8%) G trimers (Fig. 5c); the resolutions of the corresponding subtomogram averages are 19.6 Å, 20.9 Å, 18.4 Å and 20.9 Å, respectively (Supplementary Fig. 2c). The orthogonal sectional views of the super-complexes clearly show that the central G trimer contacts its neighboring G trimers (Fig. 5c, second row). The superposition of the super-complex density map with atomic models reveals possible contact sites at the N-terminus (residues 8–14) of each protomer (Supplementary Fig. 2d, e). The distances between neighboring G trimers calculated from the center coordinates of all aligned subtomograms peaked sharply at 67.9 Å in the histogram (Fig. 5d), corresponding to the side length of both hexagons and pentagons

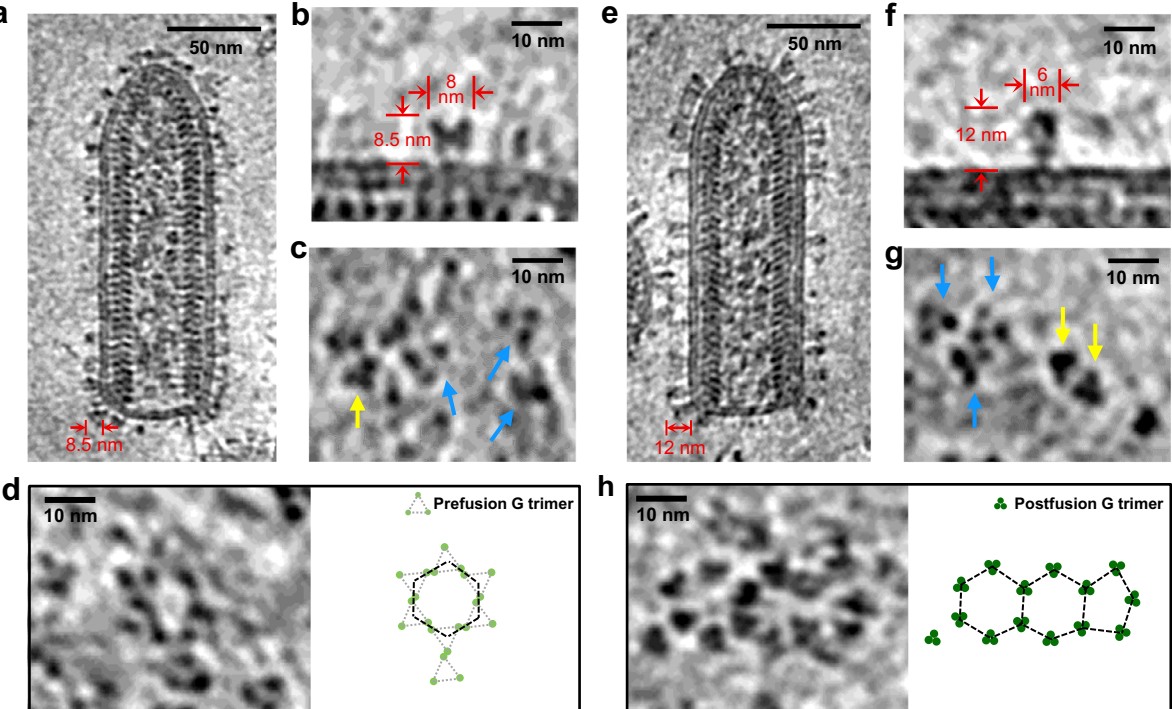

**Fig. 4 | In situ structures of VSV G trimers and their hexagonal/pentagonal distribution. a** A 6 Å-thick density slice from a reconstructed tomogram showing a representative VSV virion purified without a density gradient centrifugation step. The 8.5 nm-thickness of glycoproteins indicates that most G are in prefusion conformations. **b, c** VSV G trimers in prefusion mostly or postfusion conformation occasionally. **b** The side view of a representative G trimer in prefusion conformation from 4452 trimers obtained by template matching; **c** The top views of G trimers. Trimers in prefusion conformation, are pointed by blue arrows, while those in postfusion conformation by yellow arrows. **d** Example of a hexagonal tile formed by prefusion G trimers on the viral membrane. **e** A 6 Å-thick density slice from a reconstructed tomogram showing a representative VSV virion purified with a density gradient centrifugation step. The 12 nm-thickness of glycoproteins indicates that most G are in postfusion conformations. **f, g** VSV G trimers in postfusion mostly or prefusion conformation occasionally. **f** The side view of a representative G trimer in postfusion conformation from 6030 trimers obtained by template matching; **g** The top views of G trimers. Trimers in postfusion conformation, are pointed by yellow arrows, while those in prefusion conformation by blue arrows. **h** Two connected hexagonal tiles joined to a pentagonal tile formed by postfusion G trimers on the viral membrane.

formed by the postfusion G trimers (Fig. 4h). Interestingly, we also observed a second peak distance at 113.4 Å (Fig. 5d), which is longer than the distance of the diagonal in the postfusion pentagon (i.e., 109.8 Å) but shorter than the distance of the short diagonal (117.6 Å) in the postfusion hexagon tile (Fig. 5d, inset). In addition, this second peak of the distance histogram is flatter than the first peak (Fig. 5d), suggesting that it is a mixture of the above two diagonal distances of the pentagon and hexagon.

The hexagon tiles formed by prefusion G trimers on the viral envelope (Fig. 4d) are reminiscent of the crystalline pattern of the prefusion G trimers[19,23], where each vertex of a hexagon is occupied by one prefusion G trimer (Supplementary Fig. 3a, c). In the crystalline lattice, each G trimer is shared by three neighboring hexagons (Supplementary Fig. 3a, c). The side length of the hexagon, i.e. the distance between the centers of neighboring trimers, is 69 Å (Supplementary Fig. 3c), which is about the distance observed in situ among neighboring G trimers (Fig. 5d). Prefusion G trimer has a tripod shape and each tripod leg corresponds to domain IV (DIV) of each protomer with its fusion loops pointing towards the viral membrane (Supplementary Fig. 3b). Due to the tripod architecture, DIV in one G trimer, as illustrated in Supplementary Fig. 3b, would join DIV from a neighboring G trimer to form a DIV pair. The distance between neighboring DIV pairs in the G trimer lattice is 58 Å (Supplementary Fig. 3d).

Because the N, IM and OM layer cylinders are coaxial and the ratio of N, IM and OM sites is 1:1:1, the lateral distances between neighboring N-N, IM-IM, OM-OM sites are proportional to the radii of the N, IM and OM cylinders as illustrated by two radial projection lines normal to the axis (Fig. 5e). When viewed radially

from inside cylinder out, these sites form elongated hexagons of decreasing degrees of slenderness, leading to an equilateral hexagon with side length of 57.1 Å for the G layer at the viral envelope (Fig. 5f). This side length is about the same as the distance between the neighboring DIV pairs in the crystalline lattice of prefusion G trimers (58 Å, Supplementary Fig. 3d). This correspondence of distances indicates a matching between the hexagonal pattern formed by prefusion G trimer outside the viral membrane and the lattice formed by the OM sites inside. Such matching would enable three endodomains from a prefusion G trimer to reach out to three OM sites spanning two turns (Fig. 5g, supplementary Fig 3d, e), and each OM site to accommodate the endodomain from a neighboring G trimer (Fig. 5g). This G endodomain-OM association is expected to break due to coalesce of the transmembrane regions of G trimer during conversion from prefusion to postfusion conformation (Fig. 5h), potentially facilitating the uncoating of membrane during viral entry.

## Discussion

In this study, we have used helical cryoEM sub-particle reconstruction and cryoET subtomogram averaging to determine the near-atomic resolution in situ structures of M and N of VSV, as well as the distribution of G trimers on the VSV membrane envelope. These in situ structures establish the mode of molecular interactions among genomic RNA, N, M and G. Aside from the significance embodied in these long sought-after structures are two transformational discoveries. First, the stoichiometry between N and M sites is not the previously thought 1:1, but 1:2; a double-layer of M, composed of IM and OM,

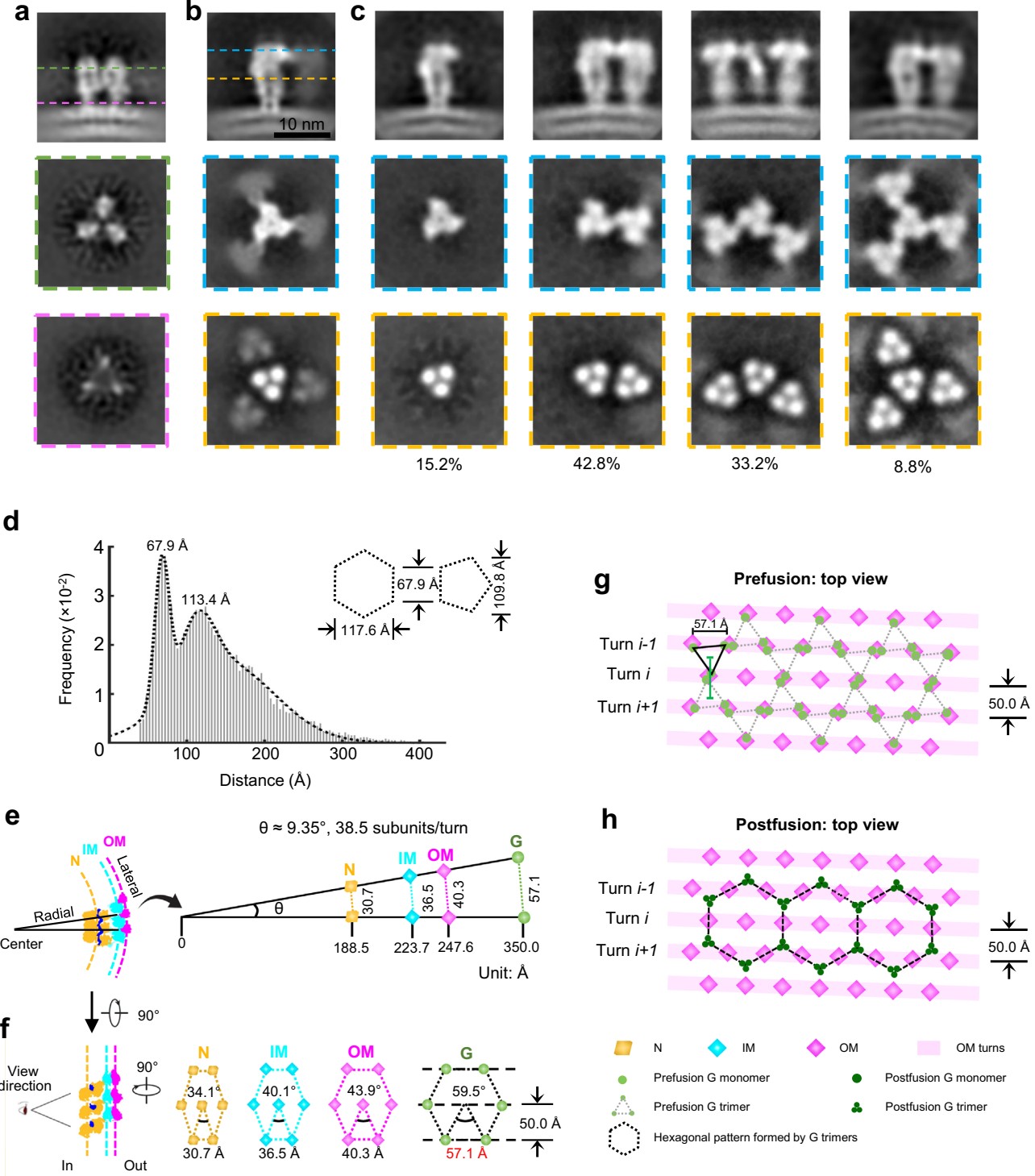

**Fig. 5 | The super-complexes of G trimers visualized by subtomogram averaging and the relationship between the spatial organization of OM and G.**
**a** Averaged density maps of G trimer in prefusion conformation. Images in the second and third row are orthogonal slice views; locations are indicated by green and magenta dashed lines in the first image. No adjacent densities show in the orthogonal views. **b, c** Averaged density maps of G trimer in postfusion conformation before classification and G trimers classified by different numbers of neighboring G trimers (super-complexes). **b** The averaged density map of all subtomograms. **c** The averaged density maps of G trimer with 0 (first column), 1 (second column), 2 (third column) and 3 (fourth column) neighboring G trimers. Images in the first row are side views of the averages; images in the second and third row are orthogonal slice views; locations are indicated by blue and golden dashed lines in the first image.
**d** Distribution of the distances of the first five nearest neighbors for each G trimer.

The dashed curve shows the fitting of a three-term Gaussian model. The second peak (113.4 Å) is between the widths of hexagon (117.6 Å) and pentagon (109.8 Å), which have an identical side length (67.9 Å, the first peak), indicating a mixture of hexagonal and pentagonal tiles. Distances between neighboring subunits (**e**) and different hexagons at different radial positions based on geometric consideration (**f**). Each dot on a hexagon represents the intersection point on the N, IM or OM cylinder by the radial projection line connecting the center of the VSV cylinder to a G subunit on the VSV envelope. Note that the indicated distance between G subunits matches those observed in our tomogram shown in (Fig. 4d) and in the crystal structure (Supplementary Fig. 3d). **g, h** Hexagonal arrangement of G trimers in both prefusion and postfusion conformations. The distances between OMs underneath the membrane match the distances between neighboring DIV pairs of prefusion G trimers outside the membrane.

surrounds a layer of N. Second, the G trimers on the viral membrane envelope are organized in pseudo-crystalline lattices, with potential tenuous interactions with OM through their endodomains.

NSVs all contain structural proteins N and M. A careful comparison of subunit interactions in the nucleocapsid of VSV with those of MeV indicates that their respective shapes—the characteristic bullet shape of VSV and the helical shape of MeV—are influenced by N-N interactions (Supplementary Fig. 4). In MeV, the assembly of the helical nucleocapsid relies entirely on N-N interactions, both vertically and laterally[24]. Each N subunit interacts directly with an adjacent N by an α helix of the N-arm (Supplementary Fig. 4, right), which is the configuration that gives rise to its slimmer helical nucleocapsid. In VSV, however, the flexible β-strand hairpin of the N-arm from $N_{U-1}$ anchors on the C-loop of $N_{U+1}$ (Fig. 3b); the flexibility of the involved N-arm and C-loop permits the different curvatures of N helical turns from the tip to the trunk (Fig. 3a, b). This flexibility also explains how different numbers of asymmetric units per helical turn are present in different VSV virions. In addition to the interactions between N subunits, other interactions, including RNA-N, N-IM, and IM-OM, make the large (~700 Å diameter), three-layered nucleocapsid sufficiently stable for packaging numerous polymerase complexes. Notably, the vertical interactions between N subunits in VSV are minimal: only IM subunits from successive turns interact with each other through a flexible N-terminal segment. The IM helix directly interacts with the N helix, with each IM positioned between two helical turns of N and stabilizing the overall structure.

One of the primary functions of M is to drive virus budding. A network of M molecules is associated with the inner leaflet of the virus envelope and the endodomains of viral glycoproteins, as shown for Newcastle disease virus (NDV)[25], MeV[26], Ebola/Marburg viruses[27], and influenza viruses[28]. However, the other primary function of M, to stabilize the nucleocapsid, is not fully understood. In the structure reported here, VSV is shown to have a double layer of M instead of a single layer, as previously thought[20,29]. Compared to the IM subunit, the OM subunit has a 40° rotation along the radial axis and an additional 40° rotation along the vertical axis (Supplementary Fig. 5b). Although IM and OM have similar structures, this large rotation of OM makes the outer-facing surfaces of IM and OM quite different (Supplementary Fig. 5c, d). Generally, the outside surface of IM is more hydrophilic, while that of OM is more hydrophobic (Supplementary Fig. 5c, d). In addition, the rotation of OM changes the interactions that are present in the IM layer (Fig. 2i and Supplementary Fig. 1b), which allows OM subunits to be distributed differently from IM. In the cryoET density, we noticed that each OM subunit has a small region attached to the membrane envelope (Supplementary Fig. 5a, red arrows); further analysis indicated that the attachment region involves two protruding loops, Pro195 – Ser199 and Ala216 – Gly218 (Supplementary Fig. 5d, green). Viewed from the side, these two loops are in a close position to interact with the membrane. In both structures of the prefusion and postfusion G trimers[18,19], 80 amino acids of G at the C-terminal, which constitute the membrane-proximal region, the transmembrane domain and the endodomain, are removed by protease digestion. The endodomain of G contains as many as seven positively charged residues (Lys and Arg) (Supplementary Fig. 5e). Interestingly, surface potential analysis of OM shows that a negatively charged region is located in the middle of the relatively hydrophobic surface of OM (Supplementary Fig. 5d). The negatively charged region comprises a small helix Val189 – Tyr192, in which the side chain of Glu191 and the main chain contribute to the negative potential. Thus, the endodomain of G could reasonably interact with this region of OM.

The double layer of M may be the result of its dual role in virion assembly, which is also supported by other studies. The OM layer in VSV is recognized to associate with the viral membrane and the endodomain of the glycoproteins, as M does in other viruses[26,30]. Membrane-derived M proteins (presumably OM) have been shown to bind nucleocapsids condensed by M proteins (presumably IM), but not intracellular nucleocapsids, which only contain RNA and N[31,32]. This binding (or lack thereof) may be explained by the different orientations, and thus the different binding affinities, of OM and IM to N. Cytosolic M protein can bind with intracellular nucleocapsids, but with much lower affinity than membrane-derived M proteins to M-condensed nucleocapsids. Without association with the M protein, the nucleocapsid was in an extended state[33,34]. The assembly of a bullet-shaped virion requires functional M proteins[35], presumably IM subunits for condensing the nucleocapsid. In other NSVs, the nucleocapsid self-assembles into a condensed helical structure ready for packaging in a virion. Their M proteins also bind the nucleocapsid prior to its incorporation into the virion[30,36–38]. Since formation of the helical nucleocapsid in other NSVs does not require association with the M protein, it is not clear if a secondary M subunit participates, in either a partial or full layer, in bridging the membrane-bound M layer and the packaged nucleocapsid in those virions.

The extensive interactions between N-IM and between IM-OM revealed by our in situ structures, as well as possible interactions between OM and prefusion G suggested by our cryoET results, suggest an alternative mechanism of VSV assembly. After genome replication, the nucleocapsid tends to assemble into a bullet-shaped structure[35,39], assisted by the association of IM. IM binding prohibits the nucleocapsid from serving as the template of viral synthesis and allows it to be transported to the plasma membrane. Meanwhile, M molecules associate with the host cell membrane[40] to form a regular mesh on the membrane that enables binding with the IM-stabilized nucleocapsid. Membrane microdomains containing G clusters that presumably share a lattice similar to that in the crystal of prefusion G trimers[19] have been observed, and the IM-decorated nucleocapsid would bind underneath such microdomains through IM-OM interactions. This association would introduce curvature to the microdomain, leading to budding of an infectious virion. Therefore, the atomic details unveiled by these in situ structures account for not only the genesis of the bullet-shaped nucleocapsid but also the recruitment of meta-stable prefusion G trimers to form a complete infectious virion. Finally, based on structural and biological results from this and other studies[41,42], we constructed a pseudo-atomic model of an entire VSV virion, which contains 1235 N, 1235 IM, 1235 OM, 11115 RNA nucleotides, 49 $LP_2$, and numerous G-trimers (Supplementary Videos 4, 5). In this model, all the available OM and IM sites are occupied though some of them might be not based on a previously reported stoichiometry of N:M of 1:1.5 (ref. 43).

From a technical point of view, our work highlights an innovative approach of sub-particle reconstruction for filamentous objects with flexible helical parameters. The workflow complements recent successes of focused refinement of RELION[44] to improve resolution of flexible local regions of a complex and sub-particle reconstruction to resolve asymmetric structures in icosahedral viruses[45,46]. This method should be generally applicable to other helical/filamentous assemblies that are common in viruses and cells.

## Methods
### Isolation of VSV virions
VSV virions were cultured as previously described[47], and full VSV particles were isolated from media by density-gradient centrifugation. Briefly, the virions were pelleted at 30,000 g for 2 h and resuspended in phosphate buffered saline (PBS, pH 7.4). The stock was then subjected to another low-speed centrifugation at 12,000 g for 5 min to remove large aggregates. The resulting suspension was loaded on a 12 ml density gradient containing 0–50% potassium tartrate and 30–0% glycerol (prepared by 6 ml 50% potassium tartrate and 6 ml 30% glycerol). After centrifugation at 40,000 g for 1 h, the VSV-containing band was extracted with a syringe and slowly diluted in 10 ml PBS. The dilution was pelleted at 30,000 g for 2 h; the

subsequent pellet was kept on wet ice for 4 h, then resuspended in a small volume of PBS prior to use for cryoEM and cryoET sample preparation.

To get virions with prefusion G trimers, the medium stock was subjected to centrifugation with 20% sucrose cushion at 30,000 $g$ for 2.5 h and resuspended in PBS. A low-speed centrifugation at 12,000 $g$ for 5 min was carried out to remove large aggregates. The resulting suspension was subsequently pelleted at 100,000 $g$ for 10 min , then resuspended in a small volume of PBS and directly used for cryoET sample preparation.

## CryoEM sample preparation, movies acquisition and drift correction

For single-particle cryoEM, each aliquot of 2.5 µl of the purified sample was applied onto a glow-discharged holey copper grid (300 mesh, Quantifoil R 1.2/1.3, Ted Pella). The grid was blotted and flash-frozen in liquid ethane with an FEI Mark IV Vitrobot. An FEI TF20 cryoEM instrument was used to screen grids and optimize the conditions.

Optimized cryoEM grids were loaded into a Titan Krios 300 kV electron microscope (Thermo Fisher Scientific) equipped with a Gatan imaging filter (GIF) Quantum LS and a Gatan K2 Summit direct electron detector. Movies were acquired with Leginon[48] in super-resolution mode at a nominal magnification of 105,000x (yielding a calibrated pixel size of 0.66 Å at the specimen level). The GIF slit width was set to 20 eV. A total number of 45 frames were acquired in 9 s for each movie, giving a total dose of ~60 e$^{-}$/Å$^2$/movie.

Frames in each movie were aligned for drift correction with the graphics processing unit (GPU)-accelerated program MotionCor2 (ref. [49]). The first and last frame were discarded during drift correction. Two averaged micrographs, one with dose weighting and the other one without dose weighting, were generated for each movie after drift correction. The averaged micrographs were binned $2 \times 2$ to yield a pixel size of 1.325 Å. The micrographs without dose weighting were used for CTF estimation and particle picking, while those with dose weighting were used for particle extraction and in-depth processing.

## 2D and 3D classification

The data processing workflow for VSV cryoEM dataset is summarized in Supplementary Fig. 6. The CTF estimation of each micrograph was performed by CTFFIND4 (ref. [50]). From a total of 2008 micrographs, 1692 good ones were selected and phase-flipped by Bsoft[51]. 7,732 VSV virions (start-end coordinates pair) were then picked up manually using RELION[44].

With RELION's helical extraction function, each virion was subdivided into overlapping segments with dimensions of $800 \times 800$ square pixels and an inter-box distance of 100 Å. 97,921 particles were extracted and 2× binned to $400 \times 400$ square pixels (pixel size: 2.65 Å) to speed up further data processing. These particles were then subjected to reference-free 2D classifications. Classes with bad particles (i.e., classes with fuzzy or uninterpretable features) and those containing either the virion tip region or the bottom region were discarded, yielding a total of 72,923 selected particles.

To account for the variable numbers of asymmetric units per helical turn, we sorted particles according to their helical parameters through four rounds of 3D classification. The first three rounds generated the density maps used as the initial references, with which we subsequently sorted the particles into different classes in the fourth and final round through multi-reference 3D classification. In the first round of 3D classification, we used *relion_helix_toolbox* with known and calculated helical parameters of VSV and the atomic models of N[20] (PDB: 2WYY) and M[21] (PDB: 2W2R) to generate 6 reference models. These models ranged from 35.5 to 40.5 asymmetric units per turn with an interval of 1. To calculate the helical parameters (helical rise and helical turn) of these models, we consulted a previous study[20] to find

the distance between two adjacent turns (helical pitch), which is about 50 Å. Thus, the helical rise is 50 Å/units per turn, and the helical turn is −360°/units per turn (a negative degree indicates left-handedness of the helix). Each segment was transferred to a density map (20 Å low-pass filtered) using the *molmap* command in Chimera[52], and the 6 resulting density maps were used as the initial references for 6 independent 3D classification jobs. In each job, all good particles were employed and helical parameters imposed according to the number of asymmetric units per turn. 3 classes were asked for each job, and no helical parameter search was applied. The reconstructed map with the best features among the 3 classes was selected as the reference for the second round of classification; this was repeated for each job. In total, 6 maps were selected from 6 independent 3D classifications. In the second round of 3D classification, a single Star file was used, containing the 6 maps selected from the first round and their corresponding helical parameters as the initial references. All good particles were employed again, resulting in 6 classes and 6 corresponding maps. In the third round, each class, containing a particle subset and map, from the second round was selected and subjected to an independent 3D classification job. For each job, 3 classes were asked and helical parameters were searched for within a very small range. Thus, the particles were further classified, leading to classes with improved reconstructed maps and more accurate helical parameters. However, for the job related to 40.5 asymmetric units per turn, we noticed that one of the 3 classes had much lower resolution compared to the other two, suggesting the existence of particles with 41.5 asymmetric units per turn. Thus, we performed additional 3D classification jobs similar to those in the second round for this particle subset, resulting in a reconstructed map of and refined helical parameters for particles with 41.5 asymmetric units per turn. Through the three rounds of classifications above, 7 good initial reference maps and their corresponding helical parameters were prepared. In the fourth round, these improved maps and helical parameters were used as references for a multi-reference 3D classification job. All good particles were employed, and 7 classes with variable numbers of asymmetric units per turn (35.5-41.5) were identified, with particle distribution percentages of 6.8%, 15.1%, 12.6%, 31.8%, 20.7%, 6.0% and 7.0%, respectively (Supplementary Fig. 6 and Supplementary Table 2).

## Helical sub-particle reconstruction

3D classification resulted in 7 classes with different helical parameters; however, further 3D auto-refinement of each class failed to achieve high resolution, which indicated local flexibility inside the helix. Subsequent data processing focused on the major class, which has 38.5 asymmetric units per turn and comprises 31.8% of the total number of particles. To accommodate the helix's flexibility and improve resolution, we re-extracted small patches from the helical segments as sub-particles (Supplementary Fig. 6) with a custom-designed program, called *helisub.C*[53]. This program locates the position of each sub-particle according to the helical parameters from 3D classification. It takes the STAR file of the selected class, and the refined helical parameters (helical rise: 1.30 Å, helical turn: −9.35°), as well as the radius (210 Å) of the helical segment, as input. To avoid duplicates, we extracted only 60 sub-particles from each segment's central helical turns. In total, 1,392,420 sub-particles were extracted from 23,207 maternal segments, with a box size of $160 \times 160$ (unbinned). Along with the image stack of all the sub-particles, a new data STAR file, containing the Euler angles and offsets of each sub-particle, was also generated by *helisub.C*. The sub-particles in this image stack and the STAR file were used as input for 3D auto-refinement (local search) RELION, yielding a density map at 3.47 Å resolution (Supplementary Fig. 7). The initial reference for this auto-refinement step was directly obtained from the image stack of the sub-particles utilizing the *relion_reconstruct* command.

The effective resolution was estimated based on the "gold standard" refinement procedures and the 0.143 Fourier shell correlation

(FSC) criterion (Supplementary Fig. 7b). Local resolution was estimated using Resmap[54] (Supplementary Fig. 7c).

## Atomic modeling and model refinement

Atomic model building was accomplished in an iterative process involving Chimera[52], Coot[55] and Phenix[56]. First, the crystal structures of VSV matrix Protein M[21] (PDB: 2W2R) and nucleoprotein N[17] (PDB: 2GIC) were fitted into the cryoEM map by Chimera. Fitting revealed an extra density corresponding to single stranded RNA, whose model was subsequently built manually with Coot. This initial model was then refined using Phenix in real space with secondary structure and geometry restraints. Subsequently, the resulting model was manually adjusted for with Coot by fixing those residues not matching the cryoEM density. The Phenix and manual refinements were repeated until no improvement was possible. Refinement statistics of the VSV nucleocapsid are summarized in Supplementary Table 3. The model was also evaluated based on MolProbity scores[57] and Ramachandran plots (Supplementary Table 3).

## CryoET sample preparation, movies acquisition and drift correction

For cryoET, two batches of grids were prepared to get G in both prefusion and postfusion conformations. Fiducial gold beads of 5 nm/10 nm size were added into the purified VSV sample at a ratio of 1:20 (V/V). Each aliquot of 2.5 μl of the sample was applied onto a glow-discharged holey copper grid (200 mesh, Quantifoil R 3.5/1, Ted Pella). The grid was blotted and flash-frozen in liquid ethane with an FEI Mark IV Vitrobot. An FEI TF20 cryoEM instrument was used to screen grids and optimize the conditions.

Optimized cryoEM grids were loaded into the same Titan Krios 300 kV electron microscope (Thermo Fisher Scientific) equipped with a Gatan imaging filter (GIF) Quantum LS. The tomography data collection was performed with SerialEM[58] software v3.8, with a dose-symmetric scheme[59]. The beam was aligned in nano-probe mode and the GIF slit width was set to 20 eV. Two batches of tilt series were collected. The first batch was recorded with a Gatan K2 Summit direct electron detector, at a nominal magnification of 81,000X, calibrated pixel size of 1.74 Å, counting mode; the other batch with a Gatan K3 detector, at a nominal magnification of 64,000X, calibrated pixel size of 0.69 Å, super mode. Tilt series movies were all collected in dose-fraction mode with defocus range 2.5-4.0 um, tilt range −60°-60°, tilt increment 3° with constant dose, total dose ~120 e$^-$/Å$^2$. 30 tilt series and 26 tilt series were acquired for the two batches of tilt series, respectively. Frames in each movie were then aligned for drift correction with the graphics processing unit (GPU)-accelerated program MotionCor2 (ref. [49]). The averaged micrographs were used for the following tomogram reconstruction.

## Tomogram reconstruction and CTF deconvolution

The defocus value of each tilt images was estimated by CTFFIND4 (ref. [50]). Fiducial-based tilt series alignment and tomographic reconstruction were performed with IMOD[60] v4.9 software package. Gold beads were manually picked and automatically tracked. The fiducial model was corrected when the automatic tracking failed. The final alignment was computed without solving for any distortions. For each tilt series, two sets of tomograms were reconstructed. The first set was 2 times binned tomograms reconstruction by weighted back projection and used for subtomographic averaging. The second set was 4 times binned tomograms with SIRT-like filter, equivalent to those reconstructed by SIRT algorithm with 10 iterations, used for manual selection of virions' axes.

For better visualization, 4 times binned SIRT-filtered tomograms shown in Figures were CTF deconvolved using ISONET[61] with snrfalloff 0.7 and deconvstrength 1.

## Virus classification based on numbers of asymmetric units per helical turn

Due to the heterogeneity of VSV particles, virions were grouped into 7 classes according to their number of asymmetric units per turn. For the tomograms reconstructed from the first batch of tilt series, we manually selected 124 virions, and each of them was registered by a vector pointing from the blunt end to the tapered end. To simplify subsequent processing, only their trunk regions were taken into consideration in the following. The helical parameter solved from the previous cryoEM helical reconstruction consists of 37.5 asymmetric units per turn and a 50.8 Å rise along the helical axis[20]. The seven sets of helical parameters we used for virus sorting have identical helical rise, but the number of asymmetric units per turn ranges from 35.5 to 41.5. For each trunk, seven left-handed helical samplings, each with a different units-per-turn measurement, were applied to generated initial positions and orientations on the cylindrical lattice, which were then used to extract subtomograms. With these defined initial orientations, seven reconstructions were obtained after averaging without search. Among the seven reconstructions, only the one which coincided with the helical parameter set of each individual virion showed the strongest features, while others were smeared due to mismatching. The class which had 38.5 asymmetric units per turn comprised the majority, with 57 out of 124 virions belonging to this class. For the second batch of tomograms, 39 out of 103 virions belonged to the major class with the same strategy. Thus, virions with 38.5 asymmetric units per turn were considered in the following particle picking and subtomogram averaging.

## Particle picking (G trimer) for subtomogram averaging

The averaged structure of G trimers in postfusion conformation[62] was low-pass filtered to 40 Å and used as initial template. 4x binned tomograms were cropped into sub-regions, with each sub-region containing only one virion. Template matching was performed with *templateSearch* function in emClarity[63]. For each sub-region, 400 highest-scoring cross-correlation peaks were extracted, with a minimal distance of 54 Å (radius of G in postfusion conformation) between peaks to prevent multiple detections. To remove obvious false positives, such as the fiducial gold beads and the M proteins inside the virions, geometric constraints based on the bullet (cylindrical) shape of the virions were applied to exclude the peaks which were not on the viral membrane. 9443 subtomograms from 57 virions were retained and then refined in Relion[64].

We then extracted subtomograms with a box size of 180 from 2 times binned tomograms with the coordinates of subtomograms derived from template matching. The orientation of each subtomograms has three Euler angles calculated from the reported orientations of template match and denoted as parameters within the Relion Star file: rot (·rlnAngleRot), tilt (·rlnAngleTilt) and psi (·rlnAnglePsi). With the pre-determined orientations, an initial reference was obtained by *relion_reconstruct*. The refinement converged into a density map of G in the postfusion conformation of ~15 Å. In order to further remove false positives, especially the subtomograms containing only the membrane but no G, a skip-align classification was performed with a tight mask round the ectodomain of G. Two classes were claimed here, and the class without the ectodomain of G was excluded. Eventually, 6030 subtomograms from 57 virions, which have 38.5 asymmetric units per helical turn, were retained and used in the following analysis.

For the second batch of tomograms, a density map of prefusion G trimer was generated from the model of crystal structure by Chimera molmap function and then filtered to 40 Å as initial template. With the same strategy, 5054 subtomograms from 39 virions were obtained from template matching and refined in Relion. After false positives removal, 4452 subtomograms were contributed to the final averaged density map.

## Subtomographic averaging and 3D classification

As for the subtomograms averaging of G, local searches from auto-sampling of 1.8 degrees and 4 pixels offset searching range were performed by Relion[44], with three-fold symmetry imposed during the refinement. The final resolution was estimated with two independently refined maps from two halves of the dataset by *relion_postprocess* and determined to be 14.9 Å and 15.8 Å, respectively, at 0.143 criteria with gold standard FSC.

For the averaging of postfusion G trimers with different numbers of neighbors, the workflow is summarized in Supplementary Fig. 2. A new STAR file was generated after C3 symmetry was released by *relion_particle_symmetry_expand* and 3 times of numbers of subtomograms were subject to a skip-aligned 3D classification, with a mask around one of the three neighbors (Supplementary Fig. 2a, b). Two classes were claimed here. Based on how many symmetrized copies belonged to the first class (with neighboring G protein), all original subtomograms were divided into 4 groups: subtomograms whose 0, 1, 2, 3 symmetrized copies went to the first class originally had 0, 1, 2, 3 neighbors adjacent to the central G protein, respectively (Supplementary Fig. 2a, b). 4 groups of G trimers with 0, 1, 2 and 3 neighbors contained 918 (15.2%), 2581 (42.8%), 2004 (33.2%) and 527 (8.8%) subtomograms, respectively. The groups were then refined individually, with C3 symmetry imposed for groups of 0 and 3 neighbors and C1 for groups of 1 and 2 neighbors. The final resolutions of these 4 averaged maps were also estimated by *relion_postprocess* and determined to be 18.4 Å, 19.6 Å, 20.9 Å and 20.9 Å at 0.143 criteria with gold-standard FSC (Supplementary Fig. 2c).

## Reconstruction of an entire bullet-shaped virion structure

We divided the nucleocapsid of the virion into tip and trunk regions and reconstructed the trunk region of the virion first. The trunk region is a regular helix, with the same diameter and helical parameters per turn. Based on the asymmetric unit structure (1 N, 1 IM, 1OM and 9 nucleotides) described above and the known helical parameters of a 38.5 subunit/turn helix (helical twist = −9.35°, helical rise=1.30 Å), a reconstructed model of 120 asymmetric units was constructed by RELION[44] (*relion_helix_toolbox*). The model fit perfectly into the cryo EM density of the major class from helical sub-particle reconstruction (class 4, 38.5 subunits/turn) (Supplementary Fig. 8a).

Next, we reconstructed the tip region. To find the number of subunits per turn and the subunits' relative orientation in the tip, we manually picked 9515 tips from all micrographs and carried out a 2D classification (Supplementary Fig. 8b). Based on previous studies[20] and the 2D classification data, we concluded that the tip region is composed of a spiral whose diameters increase turn by turn from the pole. In addition, the angles between the long axis of the asymmetric unit in the spiral and the axis of the trunk increase gradually as the spiral proceeds from the pole (Supplementary Fig. 8c). Remarkably, we noticed that the distance between turns in the tip region remains the same (~50 Å). To simplify the geometric model of the tip region, we treated each turn as a standard helical turn.

As we already knew that each asymmetric unit corresponds to 9 RNA nucleotides and the RNA strand is uninterrupted, we used the RNA strand as a good reference to calculate the diameter of each turn in the tip and subsequently the number of asymmetric units. We projected the major class from 3D classification of the trunk (including two membrane layers) and the aforementioned trunk model (Supplementary Fig. 8c) to get the side view of the trunk. By aligning the projection images of the trunk and the 2D classification of the tip, we were able to locate the position of the RNA strand inside the 2D classification of the tip (Supplementary Fig. 8c, yellow dashed lines). Subsequently, we sketched the membrane curve and defined the edges of the RNA strands by translating the curves to both sides

(Supplementary Fig. 9a). We then measured the diameters of the RNA and calculated the subunit numbers of each turn (Supplementary Fig. 9a and Supplementary Table 4). Further analysis showed that the diameter of each turn increases from the pole, while the size of that increase decreases. With the distance between adjacent turns and the difference in diameter between adjacent turns, we could calculate the slope of tangent line, which indicates the tilt angle of subunits in each turn except for the first turn (Supplementary Fig. 9b, and Supplementary Table 4). Thus, we were able to reconstruct each turn of the tip region. Using these turns, as well as manual refinement, we reconstructed the full tip region, which includes 9 turns. Together with the first turn of the trunk, these 10 turns contain 291 subunits (Supplementary Fig. 9c). Eventually, we joined the trunk region and tip region together as a full length nucleocapsid model containing 1235 subunits; each subunit contains 1 N, 1 IM, 1 OM and 9 RNA nucleotides.

Placement of $LP_2$ inside the nucleocapsid and G trimers on the envelope are based on the cryoET reconstruction in this study and previous study[41]. The relative spatial positions between LP2 and adjacent N molecules documented in this previous study were used to place a total of 49 LP2 molecules in the virion model. So far, we built a pseudo-atomic model of the entire virion (Supplementary Video 4). The PDB file and associated MRC density maps, together with a Chimera[52] session (.py file) to display them, are publicly available from the corresponding author.

## Graphics visualization

Visualization of the density maps, atomic models, figures and movies, are made by UCSF Chimera[52] and IMOD[60].

## Reporting summary

Further information on research design is available in the Nature Research Reporting Summary linked to this article.

# Data availability

The sub-particle reconstruction density map generated in this study has been deposited in the Electron Microscopy Data Bank under accession code EMD-26841, the corresponding atomic model is deposited in the Protein Data Bank under the accession code 7UWS. The previously published VSV structures used in this study are available in the Protein Data Bank under accession codes 2GIC, 2WYY and 2W2R. The pseudo-atomic model of the entire VSV virion generated in this study is publicly available from the corresponding author.

# Code availability

The program *helisub.C* is provided in Github (https://github.com/EICN-UCLA/helisub).

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

## Acknowledgements

We thank Titania Nguyen for editing the manuscript. This project was supported in part by grants from the US NIH (GM071940 and AI094386 to Z.H.Z., AI171426 to M.L.), a UCLA seed grant by philanthropists Steven and Laurie Gordon (to Z.H.Z.) and a grant from University of Science and Technology of China (the Fundamental Research Funds for the Central Universities WK9100000025 to K.Z.). We acknowledge the use of resources at the Electron Imaging Center for Nanomachines supported by UCLA and grants from the NIH (1S10OD018111 and 1U24GM116792) and the National Science Foundation (DBI-1338135 and DMR-1548924).

## Author contributions

Z.H.Z., M.L. and P.G. conceived the project. J.T. cultured VSV with the supervision of M.L. K.Z. and Z.S. purified VSV and prepared cryoEM and cryoET grids. K.Z. recorded cryoEM images. K.Z. and Z.S. recorded cryoET tilt series. P.G. developed the helical sub-particle reconstruction method and worked together with K.Z. to process the cryoEM data. K.Z. built atomic models. Z.S. performed cryoET reconstructions and sub-tomogram averaging. Z.H.Z., K.Z., Z.S. and M.L. interpreted results and wrote the paper; all authors edited and approved the paper.

## Competing interests

The authors declare no competing interests.
