## [Peer review file · Nature Communications]

REVIEWER COMMENTS

Reviewer #1 (Remarks to the Author):

K. Zhou et al present an updated study on the ultrastructure of the intact virion of vesicular stomatitis virus. This structure resolves many aspects of the virion initially observed in their manuscript from 2010. An improvement in microscopy technology, sorting of heterogeneous viral particles, software, and new local symmetry averaging has allowed them to reinterpret, among other things, the distribution of the matrix protein within the virion. Here it is shown, rather than a single polymerized layer of matrix protein (M), M exists in two layers which they designate as IM and OM. The second M layer accommodates a previously unresolved density in the initial structure. The manuscript describes in depth the interactions between N-RNA, N-N, N-IM, IM-IM, IM-OM and OM-OM. Additional new findings include the potential association of OM with the viral membrane, and new tomograms and density from subtomogram averaging yield placement and network association of the VSV glycoprotein (G) on the surface of the virus. G trimers cluster in varied arrangements, consistent with lattice arrangements observed in previously determined crystal structures. The manuscript is a generational jump forward on the architecture of VSV and gives useful insight on how the virion is assembled layer by layer. The methodology and findings are excellent. A few minor suggestions are made below that would improve the manuscript.

Could the authors show an orthogonal view of Supplementary Fig. 2d/e to allow the reader to assess fitting in the perpendicular direction?

Page 11, line 224: Figure 5e might be confusing to the casual reader, ie. that it is looking at local arrangements from the perspective of looking from the outside of the virion. Could the authors give a little more detail on this here or in the legend.

Page 15, line 342: Can the authors provide detail of the placement of L and P and the choice of 49 copies in the assembled model or cite their recently published paper in PNAS (e2111948119). This is cited in the Supplementary material but the connection to that study would be useful to the reader via the main manuscript.

As helisub.C is a new program developed and used for these studies, could the authors give a little more detail about what it does and how it functions?

Minor errors:

Page 2, line 4: between should perhaps be beneath

Page 3, line 29, with other occurrences including supplement: Ref. can be removed from reference designation

Page 3, line: 13 Å resolution, is this the correct resolution therein

Reviewer #2 (Remarks to the Author):

Atomic model of VSV and mechanism of assembly, Zhou, Si, Ge, Tsao, Luo and Zhou (2022)

I have two problems with this paper.

1. The first part of the paper shows the structure of the nucleocapsid of the virus and two layers M by cryoEM. I think this part is fine but these authors have just published in PNAS about this same structure but done by cryoEM tomography, also with two layers on M, but with a lower resolution. I think that the new data in this manuscript is very nice but the authors should mention that they write a paper already in PNAS, so, in a way, although the resolution is better, the data are not really new!

2. The tomography on the spikes is really strange. The authors say that the virus was at pH 7.4 but the pictures on the spikes on the virus particle are at low pH. They mention that they have pre- and post-fusion structure of G but the “pseudo-crystalline” structure you can see only at pH 5.5! (J Cell Biol. 2010;191(1):199-210. doi:10.1083/jcb.201006116).

The particles that used for tomography have very few spikes, a lot less than you see on figure Suppl. Fig 8, probably the virus that used for the tomography was changed, low pH and probably the particles have lost the spikes. Also, when the spikes become at low pH, they will change the membrane as well and probably also the layer of M! This part of the paper cannot be published in this way!

Reviewer #3 (Remarks to the Author):

Vesicular stomatitis virus (VSV) is a widely studied prototype for viruses that have negative sense RNA. It is a member of the rhabdovirus family, as is rabies virus and other human, animal, and plant pathogens. One of the properties of rhabdoviruses that distinguishes them from other negative sense RNA viruses is their bullet-like shape, in which the core (nucleocapsid) is wound into a helical structure throughout much of its length. The helical symmetry enabled determination of the high resolution structure of the virus particle by cryo electron microscopy (cryo-EM) reported here. An earlier cryo-EM study from the same group (reference 20) determined the structure to 10.6 Å resolution (quoted here as 13 Å), whereas here the helical part of the virus was determined at 3.5 Å resolution. In addition, cryo electron tomography (cryoET) was used to analyze the organization of the envelope glycoprotein (G protein).

Noteworthy results that had not been documented previously either from the 10.6 Å structure or X-ray crystallography analysis of individual components include:

1. Flexibility in the number of subunits per turn of the helix, ranging from 35.5 to 41.5 subunits per turn. This is likely an important feature of the biology of the virus, in which particles of different lengths and compositions are assembled under different conditions. This flexibility degraded the resolution obtained by helical averaging over entire particles, which was addressed by sorting helical segments and analyzing by sub-particle reconstruction, which is described in the methods.
2. The basic subunit consists of 9 nucleotides of RNA encapsidated by a single copy of the nucleocapsid (N) protein and two copies of matrix (M) protein. The N protein-RNA complex was generally consistent with previous data from X-ray crystallography, although several important differences, including those that contribute to flexibility in the helix, were documented.
3. The most surprising result is that there are two layers of M protein, an inner layer associated with N protein (IM) and an outer layer (OM) associated with the virus membrane (envelope). Previous data, including the proposed fit to the 10.6 Å structure, had presumed that there was only a single M protein layer. It has been proposed previously that there might be two populations of M protein, one that interacts with nucleocapsid and another that interacts with membranes, but the structure determined here was quite unexpected (at least by this reviewer).
4. Subtomogram averaging provided new information about the arrangement of G protein in the envelope. In particular, pseudo-crystalline arrays of G protein with predominantly hexagonal and occasionally pentagonal tiles were observed. This may reflect the interaction of the G protein endodomain with the underlying M protein layer.

Several problems with the manuscript detract from its overall high impact:

1. Prominent among these is the fact that in the virion, N protein and M protein have a stoichiometry of 1:1.5, not 1:2. This was originally documented most convincingly by a scanning transmission electron microscopy study (Thomas et al. *J. Virol.* 54, 598-607 (1985)), and has been documented by a variety of analytical techniques since then. This ratio has been remarkably consistently observed among a wide variety of experimental conditions, including many biological and genetic manipulations. Thus line 9 in the abstract, lines 246 and 324 in the Discussion, and line 535 in the legend to Figure 6 are incorrect based on previous data. Due to the limitations of the helical reconstruction from cryo EM data, there are a number of possible explanations for how there can be two layers of M protein, but a ratio of M to N of 1.5, but this needs to be addressed.

2. Figure 6 does not add much to the manuscript. Besides being incorrect in the stoichiometry to N to M, there is no evidence for assembly of bullets in the cytoplasm as depicted in Figure 6a (quite the opposite). If the authors want to hypothesize that the IM layer forms first before the OM layer, that can be simply stated in the text, but alternative models, such as coordinate assembly of the two layers, are also possible in the absence of data to address this question. The content of Figures 6b-e is already clearly illustrated in the accompanying figures and supplemental videos.

The manuscript is well written and clearly describes the data analysis, interpretation and conclusions, all of which meet expected standards in the field, with the exceptions noted above. There is enough detail provided in the methods for the work to be reproduced.

Summary of Responses and Revision

We thank the three reviewers for their time and constructive comments, and the editor for the opportunity to respond to these comments. As you can see from our itemized responses below, we have addressed all the reviewers' criticisms thoroughly and revised the paper accordingly. In particular, to address the concern of Reviewer #2, we performed new cryoET experiment of freshly prepared VSV virion without high-speed density gradient and captured more prefusion conformation. The new results with predominantly prefusion conformation also show pseudo-crystalline pattern of G trimers, thus confirming our original conclusion. Other major changes include deletion of Figure 6 as suggested, more detailed description on software, revising text to acknowledge prior work on N:M stoichiometry. To facilitate your perusal of this document, we have copied the reviewers' comments verbatim in **black** and our answers (**Ans**) are shown in **blue** in the following text.

Reviewer #1:

K. Zhou et al present an updated study on the ultrastructure of the intact virion of vesicular stomatitis virus. This structure resolves many aspects of the virion initially observed in their manuscript from 2010. An improvement in microscopy technology, sorting of heterogeneous viral particles, software, and new local symmetry averaging has allowed them to reinterpret, among other things, the distribution of the matrix protein within the virion. Here it is shown, rather than a single polymerized layer of matrix protein (M), M exists in two layers which they designate as IM and OM. The second M layer accommodates a previously unresolved density in the initial structure. The manuscript describes in depth the interactions between N-RNA, N-N, N-IM, IM-IM, IM-OM and OM-OM. Additional new finding include the potential association of OM with the viral membrane, and new tomograms and density from subtomogram averaging yield placement and network association of the VSV glycoprotein (G) on the surface of the virus. G trimers cluster in varied arrangements, consistent with lattice arrangements observed in previously determined crystal structures. The manuscript is a generational jump forward on the architecture of VSV and gives useful insight on how the virion is assembled layer by layer. The methodology and findings are excellent. A few minor suggestions are made below that would improve the manuscript.

1. Could the authors show an orthogonal view of Supplementary Fig. 2d/e to allow the reader to assess fitting in the perpendicular direction?

Ans: We now provide a side view of the trimer as required in new Supplementary Fig. 2e.

2. Page 11, line 224: Figure 5e might be confusing to the casual reader, ie. that it is looking at local arrangements from the perspective of looking from the outside of the virion. Could the authors give a little more detail on this here or in the legend?

Ans: We have modified the old Figure 5e (new Fig. 5e, f) to improve clarity. We also rewrote the related figure legend as follows:

e, f, Distances between neighboring subunits (e) and different hexagons at different radial positions based on geometric consideration (f). Each dot on a hexagon represents the intersection point on the N, IM or OM cylinder by the radial projection line connecting the center of the VSV cylinder to a G subunit on the VSV envelope. Note that the indicated distance between G subunits match those observed in our tomogram shown in (Fig. 4d) and in the crystal structure (Supplementary Fig. 3d).

3. Page 15, line 342: Can the authors provide detail of the placement of L and P and the choice of 49 copies in the assembled model or cite their recently published paper in PNAS (e2111948119)? This is cited in the Supplementary material but the connection to that study would be useful to the reader via the main manuscript.

Ans: This paper is now cited in line 320 as requested. We now also explained how we placed LP₂ in the revised Method section (lines 915-918).

4. As helisub.C is a new program developed and used for these studies, could the authors give a little more detail about what it does and how it functions?

Ans: We now added a description about the functionality and usage of *helisub.C* (lines 753-760).

5. Page 2, line 4: between should perhaps be beneath

Ans: We have revised the original sentence to avoid confusion.

6. Page3, line29, with other occurrences including supplement: Ref. can be removed from reference designation

Ans: This was to follow citation format per journal convention. We will leave this formatting style to the copy editor.

7. Page3, line: 13 Å resolution, is this the correct resolution therein?

Ans: Great catch! It should have been 10.6 Å.

Reviewer #2:

Atomic model of VSV and mechanism of assembly, Zhou, Si, Ge, Tsao, Luo and Zhou (2022)

I have two problems with this paper.

1. The first part of the paper shows the structure of the nucleocapsid of the virus and two layers M by cryoEM. I think this part is fine but these authors have just published in PNAS about this same structure but done by cryoEM tomography, also with two layers on M, but with a lower resolution. I think that the new data in this manuscript is very nice but the authors should mention that they write a paper already in PNAS, so, in a way, although the resolution is better, the data are not really new!

Ans: The two papers are on different topics. The previous paper, entitled “Locations and in situ structure of the polymerase complex inside the virion of vesicular stomatitis virus”, documents direct visualization of the **polymerase** inside the virion. The method used there is only cryo electron tomography (cryoET), thus limited in resolution to 7.5 Å. The current paper focuses on high-resolution *in situ* structures of M and N, and their organization in relation to G, and was done by cryoEM through sub-particle reconstruction in order to build atomic models. Localization of the polymerase is not the purpose of this paper. To prevent potential confusion about these different scopes of the two papers, we have deleted Figure 6 in the revised paper.

2. The tomography on the spikes is really strange. The authors say that the virus was at pH 7.4 but the pictures on the spikes on the virus particle are at low pH. They mention that they have pre- and post-fusion structure of G but the “pseudo-crystalline” structure you can this only at pH 5.5! (J Cell Biol. 2010;191(1):199-210. doi:10.1083/jcb.201006116).

The particles that used for tomography have very few spikes, a lot less that you see on figure Suppl. Fig 8, probably the virus that used for the tomography was changed, low pH and probably the particles have lost the spikes. Also, when the spikes become at low pH, they will change the membrane as well and probably also the layer of M! This part of the paper cannot be published in this way!

Ans: Your thought is a good one and that was how we initially suspected. Indeed, for that reason, we had previously repeated the same protocol to ensure neutral pH condition throughout our original experiments. The result was the same; therefore, our interpretation was that the postfusion conformation was caused by the mechanical stress (after all, the prefusion conformation of G is at a metastable state) that the glycoprotein encountered during the three rounds of the high-speed centrifugations, including one density gradient step.

To test this hypothesis, we eliminated the density gradient step in our sample isolation. With the same freezing and imaging conditions, our reconstructed tomograms indeed reveal that most of the spikes

are now in the prefusion conformation (new Fig. 4a-c). In addition, we also observed the hexagonal patterns formed by prefusion trimers (Fig. 4d), quite similar to what we proposed in our original manuscript. We are aware that some of the G trimers were lost during the purifications or freezing, leaving some enveloped areas without G decorated. The sparsely-distributed G still exhibit hexagonal or pentagonal patterns, which indicate that this “pseudo-crystalline” structure is not merely results of close package.

Concerning the quantity of the spikes on each virion, we think the observed numbers of trimeric spikes on the virion in our reconstructed tomograms and cryoEM images are about the same, though the cryoEM image gives the impression of more G. This is because that the virions used for cryoET and cryoEM data collections were prepared with the same method. The cryoEM image is a 2D projection, thus containing superpositions of the side views of all Gs on the virion. By contrast, the slice view of the tomograms only contains G in one slice of the 3D volume. The following projection images are the projections of different thickness from a same tomogram, showing the number of visible spikes increases as the thickness increases.

Reviewer #3:

Vesicular stomatitis virus (VSV) is a widely studied prototype for viruses that have negative sense RNA. It is a member of the rhabdovirus family, as is rabies virus and other human, animal, and plant pathogens. One of the properties of rhabdoviruses that distinguishes them from other negative sense RNA viruses is their bullet-like shape, in which the core (nucleocapsid) is wound into a helical structure throughout much of its length. The helical symmetry enabled determination of the high resolution structure of the virus particle by cryo electron microscopy (cryo-EM) reported here. An earlier cryo-EM study from the same group (reference 20) determined the structure to 10.6 Å resolution (quoted here as 13 Å), whereas here the helical part of the virus was determined at 3.5 Å resolution. In addition, cryo electron tomography (cryoET) was used to analyze the organization of the envelope glycoprotein (G protein).

Noteworthy results that had not been documented previously either from the 10.6 Å structure or X-ray crystallography analysis of individual components include:

1. Flexibility in the number of subunits per turn of the helix, ranging from 35.5 to 41.5 subunits per turn. This is likely an important feature of the biology of the virus, in which particles of different lengths and compositions are assembled under different conditions. This flexibility degraded the resolution obtained by helical averaging over entire particles, which was addressed by sorting helical segments and analyzing by sub-particle reconstruction, which is described in the methods.
2. The basic subunit consists of 9 nucleotides of RNA encapsidated by a single copy of the nucleocapsid (N) protein and two copies of matrix (M) protein. The N protein-RNA complex was

generally consistent with previous data from X-ray crystallography, although several important differences, including those that contribute to flexibility in the helix, were documented.

3. The most surprising result is that there are two layers of M protein, an inner layer associated with N protein (IM) and an outer layer (OM) associated with the virus membrane (envelope). Previous data, including the proposed fit to the 10.6 Å structure, had presumed that there was only a single M protein layer. It has been proposed previously that there might be two populations of M protein, one that interacts with nucleocapsid and another that interacts with membranes, but the structure determined here was quite unexpected (at least by this reviewer).

4. Subtomogram averaging provided new information about the arrangement of G protein in the envelope. In particular, pseudo-crystalline arrays of G protein with predominantly hexagonal and occasionally pentagonal tiles were observed. This may reflect the interaction of the G protein endodomain with the underlying M protein layer.

Several problems with the manuscript detract from its overall high impact:

1. Prominent among these is the fact that in the virion, N protein and M protein have a stoichiometry of 1:1.5, not 1:2. This was originally documented most convincingly by a scanning transmission electron microscopy study (Thomas et al. J. Virol. 54, 598-607 (1985)), and has been documented by a variety of analytical techniques since then. This ratio has been remarkably consistently observed among a wide variety of experimental conditions, including many biological and genetic manipulations. Thus line 9 in the abstract, lines 246 and 324 in the Discussion, and line 535 in the legend to Figure 6 are incorrect based on previous data. Due to the limitations of the helical reconstruction from cryo EM data, there are a number of possible explanations for how there can be two layers of M protein, but a ratio of M to N of 1.5, but this needs to be addressed.

Ans: Due to averaging in our sub-particle reconstruction, we can't eliminate the possibility that some of the available OM (or IM) sites are not occupied, giving rise to a total number of M in each virion to be less than 2 times of that of N. We have revised our manuscript to reflect this uncertainty by referring to M sites (lines 31, 85 and 267) and citing this previous STEM measurement (lines 344-346).

2. Figure 6 does not add much to the manuscript. Besides being incorrect in the stoichiometry to N to M, there is no evidence for assembly of bullets in the cytoplasm as depicted in Figure 6a (quite the opposite). If the authors want to hypothesize that the IM layer forms first before the OM layer, that can be simply stated in the text, but alternative models, such as coordinate assembly of the two layers, are also possible in the absence of data to address this question. The content of Figures 6b-e is already clearly illustrated in the accompanying figures and supplemental videos.

Ans: Agree. Fig. 6 has been deleted.

The manuscript is well written and clearly describes the data analysis, interpretation and conclusions, all of which meet expected standards in the field, with the exceptions noted above. There is enough detail provided in the methods for the work to be reproduced.

In summary, we have revised the manuscript by incorporating all your suggestions and addressed the issues raised. Thank you all for your support and suggestions!

REVIEWERS' COMMENTS

Reviewer #1 (Remarks to the Author):

The revision has satisfactorily addressed all issues with the prior submission.

Reviewer #2 (Remarks to the Author):

The paper has changed a lot since the first version. I like the paper a lot more now. Think you for the new figures on G in the pre- and post-structures.

Reviewer #3 (Remarks to the Author):

My comments were addressed in the revised manuscript.